**Elevated CO$_2$, increased leaf-level productivity and water-use efficiency during the early Miocene**
Tammo Reichgelt[1,2], William J. D'Andrea[1], Ailín del C. Valdivia-McCarthy[1], Bethany R.S. Fox[3], Jennifer
M. Bannister[4], John G. Conran[5], William G. Lee[6,7], Daphne E. Lee[8]
[1]Lamont-Doherty Earth Observatory, Columbia University, Palisades, New York, USA.
[2]Department of Geosciences, University of Connecticut, Storrs, Connecticut, USA.
[3]Department of Biological and Geographical Sciences, University of Huddersfield, Huddersfield, UK.
[4]Department of Botany, University of Otago, Dunedin, New Zealand.
[5]ACEBB & SGC, School of Biological Sciences, The University of Adelaide, Adelaide, Australia.
[6]Landcare Research, Dunedin, New Zealand.
[7]School of Biological Sciences, University of Auckland, Auckland, New Zealand.
[8]Department of Geology, University of Otago, Dunedin, New Zealand.
Correspondence: Tammo Reichgelt (tammo.reichgelt@uconn.edu)
**Abstract.** Rising atmospheric CO$_2$ is expected to increase global temperatures, plant water-use efficiency,
and carbon storage in the terrestrial biosphere. A CO$_2$ fertilization effect on terrestrial vegetation is
predicted to cause global greening as the potential ecospace for forests expands. However, leaf-level
fertilization effects, such as increased productivity and water-use efficiency, have not been documented
from fossil leaves in periods of heightened atmospheric CO$_2$. Here, we use leaf gas-exchange modeling
on a well-preserved fossil flora from early Miocene New Zealand, as well as two previously published
tropical floras from the same time period, to reconstruct atmospheric CO$_2$, leaf-level productivity, and
intrinsic water-use efficiency. Leaf gas-exchange rates reconstructed from early Miocene fossils which
grew at southern temperate and tropical latitudes, when global average temperatures were 5–6°C higher
than today reveal that atmospheric CO$_2$ was ~450–550 ppm. Early Miocene CO$_2$ was similar to projected
values for 2040AD, and is consistent with Earth System Sensitivity of 3–7°C to a doubling of CO$_2$. The
Southern Hemisphere temperate leaves had higher reconstructed productivity than modern analogs likely
due to a longer growing season. This higher productivity was presumably mirrored at northern temperate
latitudes as well, where a greater availability of landmass would have led to increased carbon storage in
forest biomass relative to today. Intrinsic water-use efficiency of both temperate and tropical forest trees
was high, toward the upper limit of the range for modern trees, which likely expanded the habitable range
in regions that could not support forests with high moisture demands under lower atmospheric $CO_2$.
Overall, early Miocene elevated atmospheric $CO_2$ sustained globally higher temperatures and our results
provide the first empirical evidence of concomitant enhanced intrinsic water-use efficiency, indicating a
forest fertilization effect.

**1 Introduction**
Terrestrial plants comprise 450 Gt of carbon, representing 80% of Earth's dry carbon (C) biomass (Bar-on
et al., 2018). Globally, plants draw down ~120 Gt of atmospheric C per year through photosynthesis,
representing the largest annual C flux on Earth (Beer et al., 2010). Total plant biomass is believed to be
determined in large part by atmospheric carbon dioxide concentrations ($C_a$), and it is predicted that future
increases in $C_a$ will have a three-pronged effect on the terrestrial biosphere: 1) increased global
temperatures will shift the boundaries of climate zones and thereby the potential forest expanse (Rubel
and Kottek, 2010); 2) productivity will increase because global photosynthesis is C limited and increased
$C_a$ will have a fertilization effect on the terrestrial biosphere (Zhu et al., 2016); and 3) elevated $C_a$ will
increase plant water-use efficiency and reduce the threshold for physiological drought (Cernusak, 2020),
making more land area available for biosphere expansion (Zhou et al., 2017). Plant fossils record the
effect of past changes in climate, including $CO_2$ enrichment, and thus fossil floras provide insight into
changes in the carbon cycle and their effects on the terrestrial biosphere from a natural, whole-ecosystem
perspective.
The Miocene has been considered problematic for our understanding of Earth System Sensitivity
(ESS) to $C_a$, because most proxy-based $C_a$ estimates indicate concentrations near 300 ppm (Foster et al.,
2017), close to pre-industrial values, yet global temperatures were 5–6 °C higher than modern (Hansen et
al., 2013). Enhanced radiative forcing is required to maintain such elevated early Miocene temperatures
(Herold et al., 2010; Hansen et al., 2013), and without elevated $C_a$, climate models cannot achieve such
high global temperatures in the Miocene (Henrot et al., 2010). The early Miocene also had an expanded
biosphere compared to today, including woody vegetation in locations that are currently too cold and/or
too dry for forests (e.g. Askin and Raine, 2000; Herold et al., 2010). A biosphere of the magnitude
observed in the early Miocene fossil record requires elevated temperatures and plant water-use efficiency,
suggesting $C_a$ levels higher than estimated by most existing proxy reconstructions (Herold et al., 2010;
Henrot et al., 2010). Importantly, plant-based $C_a$ reconstructions have challenged the consensus of low $C_a$
in the early Miocene (Kürschner and Kvaček, 2009; Reichgelt et al., 2016; Tesfamichael et al., 2017;
Londoño et al., 2018) and previous interpretations of alkenone-based $C_a$ proxies are being disputed
(Bolton et al., 2016; Witkowski et al., 2019).

We applied plant gas-exchange modeling (Franks et al., 2014) to fossil leaves from the early Miocene

(~23 Ma) rainforest ecosystem from southern New Zealand preserved in the Foulden Maar deposit
(Bannister et al., 2012; Reichgelt et al., 2013; Conran et al., 2014; Lee et al., 2016) to reconstruct carbon
assimilation rates ($A_n$), intrinsic water-use efficiency (iWUE; the ratio between carbon assimilation and
stomatal conductance to water), and the $C_a$ levels required to maintain these values. The same analyses
were performed on previously published leaf $\delta^{13}C$ and micromorphological measurements from two early
Miocene fossil floras from Ethiopia (Tesfamichael et al., 2017) and Panama (Londoño et al., 2018). These
two studies were incorporated here because they represent a similar time period (early Miocene) and the
authors applied the same gas-exchange methodology. Because $A_n$, iWUE, and $C_a$ are interdependent
(Farquhar et al., 1980; Drake et al., 1997), we reconstructed these variables in concert for each fossilized
leaf recovered from these forest ecosystems. This allows us to make inferences concerning carbon
availability, productivity, and water balance in the forest.

**2 Methods**

## 2.1 Site Description

Foulden Maar (Fig. 1a) is a unique *Konservat-Lagerstätte* with abundantly preserved plants and insects (Kaulfuss et al., 2015; Lee et al., 2016) in southern New Zealand (45.527°S, 170.219°E). It was formed in an ancient maar-diatreme lake (Fig. 1b) at the Oligocene/Miocene boundary (Fox et al., 2015; Kaulfuss, 2017) and consists of ~100 kyr of annually laminated diatomite (Lindqvist and Lee, 2009; Fox et al., 2016). The Foulden maar-diatreme complex is part of the larger late Oligocene – late Miocene Waipiata Volcanic Field that produced a variety of maar volcanoes and scoria cones (Németh and White, 2003). Plant fossils used in this study were collected from a ~183 m long drill core (Fig. 1c). The Lauraceae-dominated rainforest (Bannister et al., 2012) surrounding the lake grew at a paleolatitude of ~50°S (Fig. 1a). The climate was marginally subtropical with a mean annual temperature of ~18°C, similar to modern day climates at 30°S (Reichgelt et al., 2019). The length of the growing season in this climate was ~10 months, compared to 5–6 months today, as reconstructed from the surface exposure macrofossil assemblage using the Climate Leaf Analysis Multivariate Program (Reichgelt et al., 2013).

## 2.2 Fossil leaf anatomy and paleoecology

Mummified fossil leaves were extracted from turbidite deposits that occur frequently within the Foulden Maar diatomite core (Fox et al., 2015). The diatomite is loosely consolidated allowing mummified leaves to be extracted using a combination of water and scalpels. After extraction, the leaves were cut into three pieces: one for bulk $\delta^{13}C$ analysis, one for stomatal conductance measurements, and a third as a reference specimen. Leaf $\delta^{13}C$ was measured using a Costech elemental combustion system (EA) coupled to a Delta V Plus IRMS (Thermo). In order to place measured $\delta^{13}C$ on the VPDB scale, we calibrated measurements using a two-point isotope calibration based on the USGS40 and USGS41 standards. Measurement uncertainty was calculated by replicating ~15 samples two or three times and applying averaged uncertainty to the remaining leaves. For stomatal density and geometry measurements the leaves were soaked in hydrogen peroxide ($H_2O_2$) with up to 40% dilution, with tetra-sodium pyrophosphate salt crystals ($Na_4P_2O_7 \cdot 10[H_2O]$), on a boiling plate at 40–50 °C for 1–2 hours (Bannister et al., 2012). When

the adaxial and abaxial cuticle layer could be separated, the leaf layers were cleaned of mesophyll cell
debris using small paintbrushes and both layers were stained with <0.5% Crystal Violet ($C_{25}N_3H_{30}Cl$) and
mounted on glass slides with glycerin jelly. Stomatal conductance and geometry measurements were
made on pictures at 100× magnification using TSView 7.1.1.2 microscope imaging software on a Nikon
Optiphot. Leaves were often fragmented and the cuticle wrinkled because the leaves were deposited in
turbidites. Moreover, there was strong divergence in overall cell and stomatal density, because we made
measurements on all species recovered from the sediments. To avoid systematic errors arising from
wrinkled cuticle, differing leaf architecture between species, or low cell counts (Retallack and Conde,
2020), each picture was given a standard bounding box (0.3 × 0.3 mm) on which cells were counted, to
calculate stomatal density. The number of cells in each bounding box ranged from 100 – 750, strongly
dependent on species. Five to eight pictures were taken of each leaf to constrain errors in cell density.
Stomatal size measurements were made using ImageJ 1.48v software (Schneider et al., 2012).
18 distinct leaf morphotypes were identified from the Foulden Maar drill core. Descriptions and
justification for identification are found in the Supplementary Information. Species identifications are
provided, where possible, based on paleobotanical studies from the Foulden Maar surface exposures.
Known species recovered from the Foulden Maar drill core are *Litsea calicarioides* (Fig. S1a),
*Cryptocarya taieriensis* (Fig. S1b), *C. maarensis* (Fig. S1i), *Beilschmiedia otagoensis* (Fig. S2a)
(Lauraceae) (Bannister et al., 2012), *Laurelia otagoensis* (Fig. S2h) (Atherospermataceae) (Conran et al.,
2013), and *Hedycarya pluvisilva* (Fig. S2i) (Monimiaceae) (Conran et al., 2016). Otherwise, tentative
genus or family identifications are provided, or unspecified morphotypic qualifiers, for leaves that could
not be assigned a plant group. These will henceforth be referred to as "C" (Fig. S1c), cf. Myrtaceae (Fig.
S1d), cf. *Ripogonum* (Fig. S1e), cf. *Myrsine* (Fig. S1f), "H" (Fig. S1g), cf. Elaeocarpaceae/Cunoniaceae
(Fig. S1h), cf. *Dysoxylum* (Fig. S2b), cf. *Cryptocarya* (Fig. S2c), "O" (Fig. S2d), "P" (Fig. S2e), "Q"
(Fig. S2f) and cf. *Endiandra* (Fig. S2g).
We made 375 anatomical and 80 carbon isotope measurements on 72 organically preserved fossil
leaves representing the 18 species collected from the Foulden Maar deposit (Fig. 1a). The affinities of
modern living relatives of the plant types at Foulden Maar strongly suggest that during the Miocene the
site was characterized by a multi-layered closed canopy rainforest ecosystem (Reichgelt et al., 2013;
Conran et al., 2014). In order to determine atmospheric carbon ($C_a$), intrinsic water-use efficiency
(iWUE), and carbon assimilation rates ($A_n$), the ecological strategies of the individual fossil species at
Foulden Maar need to first be established (Reichgelt and D'Andrea, 2019). Understory species rarely
experience light saturation and utilize respired $CO_2$ that has already undergone isotopic fractionation;
both conditions influence gas-exchange modelling results (Royer et al., 2019). Therefore, $C_a$
reconstructed from understory species cannot be considered indicative of true global $C_a$. Here, we
determine whether a fossil leaf type was likely in the canopy or the understory, based on 1) leaf $\delta^{13}C$, 2)
leaf cell density, and 3) sinuosity of the epidermal cell walls. A large range of leaf $\delta^{13}C$ in a single species
is indicative of different levels of light saturation, which indicates that this species may preferentially
occur in the subcanopy or in the understory (Graham et al., 2014). Leaves in the canopy, experiencing
light saturation, divide epidermal cells rapidly compared to leaves in the shade, leading to high cell
densities and relatively high leaf mass per areas in sun-exposed leaves (Šantrůček et al., 2014). Finally, a
high level of anticlinal cell wall sinuosity has been interpreted as indicative of low-light conditions
(Kürschner, 1997; Bush et al., 2017). We consider these three lines of evidence occurring in concert as
indicative of a canopy or subcanopy ecological preference.

**2.3 Modelling gas-exchange**
Atmospheric carbon dioxide ($C_a$), plant photosynthesis ($A_n$), and intrinsic water-use efficiency (iWUE)
are tightly linked (Farquhar et al., 1980; Drake et al., 1997), which allows us to solve for these parameters
iteratively, through anatomical and carbon isotope ($\delta^{13}C$) measurements of the fossil leaves. The Franks et
al. (2014) gas-exchange model solves for $C_a$, by iteratively reconstructing $A_n$ and leaf conductance to
atmospheric carbon ($G_c$), using a Monte Carlo approach. This means that every $C_a$ reconstruction has an
associated $A_n$ and $G_c$ value.

$$C_a = {A_n}\Big/{G_c \times \left(1 - \frac{C_i}{C_a}\right)} \quad (1)$$


In which $C_i/C_a$ represents the ratio of intercellular carbon to atmospheric carbon, which can be
reconstructed using known leaf fractionation processes: fractionation caused by diffusion (a),
carboxylation (b), and fractionation caused by the preferential uptake of $^{12}C$ to $^{13}C$ in photosynthesis ($\Delta$),
which is also influenced by the rate at which the leaf is photosynthesizing (Farquhar et al., 1982).

$$\frac{C_i}{C_a} = \frac{\Delta - a}{b - a} \quad (2)$$


Here, a = 4.4‰ and b = 29‰ (Farquhar et al., 1982; Roeske and O'Leary, 1984). $\Delta$ can be calculated
from the $\delta^{13}C$ of the air, derived from Tipple et al. (2010) and measurements of leaf $\delta^{13}C$ (Farquhar and
Richards, 1984; Farquhar et al., 1989). Leaf and air $\delta^{13}C$ used in the Franks et al. (2014) model are
presented in Table S1.

$$\Delta = \frac{\delta^{13}C_{air} - \delta^{13}C_{leaf}}{1 + \delta^{13}C_{leaf}} \quad (3)$$


$G_c$ is determined by the maximum capacity for conductance of a leaf surface ($G_{max}$), the ratio of
operational conductance to $G_{max}$ ($\zeta$), boundary layer conductance ($G_b$), and mesophyll conductance ($G_m$)
(Franks et al., 2014).

$$G_c = \left(\frac{1}{G_b} + \frac{1}{\zeta \times G_{max}} + \frac{1}{G_m}\right)^{-1} \quad (4)$$


$G_b$, $\zeta$, and $G_m$ are all changeable under natural conditions (e.g. Schuepp, 1993; Niinemets et al., 2009;
Londoño et al., 2018) and it is highly disputed if these variables can be determined from fossil leaf
material at all (e.g. McElwain et a., 2016; Soh et al., 2017). However, we adopt a standardized approach
put forward by Franks et al. (2014) to obtain input for these variables. $G_b = 2 \pm 0.1$ mol m$^{-2}$ s$^{-1}$, $\zeta = 0.2 \pm$
$0.02$ (Franks et al., 2009; Dow et al., 2014), and $G_m$ is determined using an empirical calibration (Evans
and Von Caemmerer, 1996).

$$G_m = 0.013 \times A_n \quad (5)$$

$G_{max}$ is determined using predominantly measurable anatomical features of the fossil leaf cuticle (Franks
and Beerling, 2009): stomatal density (SD), maximum aperture surface area ($a_{max}$), pore depth ($p_d$), and
the ratio of diffusivity of $CO_2$ in air over the molar volume of air (d/v), here taken as 0.000714 mol m$^{-1}$ s$^{-1}$
(Marrero and Mason, 1972).

$$G_{max} = {d}/{v} \times SD \times \frac{a_{max}}{p_d + \frac{\pi}{2}\sqrt{a_{max}/\pi}} \quad (6)$$

In this equation, SD can be measured directly from the leaf, $p_d$ is assumed to be the same as guard cell
width (gcw), and $a_{max}$ is determined assuming a circular opening for the aperture, with the measurable
pore length ($p_l$) as the diameter (Franks et al., 2014).

$$a_{max} = \pi \times \frac{p_l}{4} \quad (7)$$

Measurements of SD, gcw and $p_l$ used in the Franks et al. (2014) gas-exchange model are presented in
Table S1.

Because $A_n$ is required to solve $G_m$, $G_c$ is solved iteratively, though $G_c$ is largely determined by
measurable anatomical parameters. However, $A_n$ is also solved iteratively, as it is dependent on $C_a$ and the
carbon saturation value ($\Gamma$), set at 40 ppm (Franks et al., 2013).

$$A_n \approx A_0 \times \frac{(C_a - \Gamma) \times (C_{a0} + 2\Gamma)}{(C_a + 2\Gamma) \times (C_{a0} - \Gamma)} \ (8)$$


In which $A_0$ is the photosynthetic rate of a modern model species that can represent the fossil species'
photosynthetic rate, and $C_{a0}$ is the atmospheric carbon dioxide level at which $A_0$ was measured. $A_0$ for
each fossil species was derived from the compilation of photosynthetic rates presented in Reichgelt and
D'Andrea (2019). For fossil leaves with known modern relatives, we constrained the possible $A_0$ range by
only including modern relatives within the same family or order, i.e. Lauraceae for *Litsea calicarioides*,
*Cryptocarya taieriensis*, *C. maarensis*, cf. *Cryptocarya*, *Beilschmiedia otagoensis* and cf. *Endiandra*,
Myrtaceae for cf. Myrtaceae, Liliales for cf. *Ripogonum*, Primulaceae for cf. *Myrsine*, Elaeocarpaceae and
Cunoniaceae for cf. Elaeocarpaceae/Cunoniaceae, Meliaceae for cf. Meliaceae, Atherospermataceae for
*Laurelia otagoensis,* and Laurales for *Hedycarya pluvisilva*. Then, following the method of constraining
$A_0$ of modern living relatives presented in Reichgelt and D'Andrea (2019), only $A_0$ values of plants with
similar growth forms to the fossil plants, and growing in similar light environments as Foulden Maar were
included. $A_0$ and $C_{a0}$ used in the Franks et al. (2014) model, and associated ecology of fossil leaf types is
shown in Table S2.

The Franks et al. (2014) gas-exchange model thus iteratively solves for $C_a$, $A_n$, and $G_c$. However, only
leaves derived from canopy trees are likely to represent these values at light saturation. Moreover, plants
in the understory assimilate a mix of atmospheric and respired $CO_2$, which has already undergone
fractionation processes, making the calculated $C_i/C_a$ problematic. Therefore, we present the results for $C_a$,
$A_n$, and $G_c$ of leaf types most likely to be derived from canopy trees separately, as they are more likely to
not have a systematic skew.
iWUE is defined as the ratio between $A_n$ and stomatal conductance to water (Feng, 1999).

$$iWUE = \frac{A_n}{G_w} \quad (9)$$


Due to the different rates at which carbon dioxide and water vapor diffuse in air, a transformation of $G_c$ is
required to calculate $G_w$.

$$G_w = 1.6 \times G_c \quad (10)$$


Finally, cumulative annual carbon uptake through photosynthesis ($A_{tot}$) can be calculated in gC m$^{-2}$ yr$^{-1}$,
by transferring from moles to grams, including a measure for the relative time the leaf is assimilating
carbon ($\zeta$), and the length of the growing season.

$$A_{tot} = (2.6 \times \zeta \times A_n \times t_g) \times 12 \quad (11)$$


In which $t_g$ is the length of the growing season in months, which we can derive from the fossil plant
assemblage (Reichgelt et al., 2019), using the method of Spicer et al. (2009). $G_w$, $A_{tot}$, and iWUE values
for *Litsea calicarioides*, *Cryptocarya taieriensis*, *C. maarensis*, cf. Elaeocarpaceae/Cunoniaceae, and cf.
Myrtaceae are presented in Table S3. The modern reference $A_n$ and $G_w$ data are derived from Maire et al.
(2015), which, using transform functions 9 and 11, we also used to calculate iWUE and $A_{tot}$.

**2.4 Comparison to Earth System Sensitivity**
Earth System Sensitivity to $C_a$ (ESS) is the amount of temperature increase expected under a doubling of
atmospheric $CO_2$. This sensitivity is likely not static in Earth's history and is dependent on, among other
aspects, continental configuration and ocean circulation patterns (Royer, 2016). Here, we estimate global
surface temperature for the early Miocene following the approach of Hansen et al. (2013). We then use
these temperature estimates along with a broad range of commonly cited Neogene ESS, of 3–7°C
(Hansen et al., 2013; Royer, 2016), to provide a model for the expected early Miocene $C_a$. Following this
model, ultimately means that a doubling of $C_a$ compared to pre-industrial levels is expected when an
increase of global average surface temperatures ($T_s$) of 3–7°C compared to modern occurs.

Compiled deep-sea benthic foraminifera $\delta^{18}O$ data of the last 30 million years (Zachos et al., 2001)

were averaged into 20 kyr time bins. Deep-sea temperatures ($T_d$) were then calculated using the linear
transfer functions of Hansen et al. (2013), which depend on the presence of sea-ice.

$$T_d = 5 - 8 \times \frac{\delta^{18}O - 1.75}{3} \ IF \ (\delta^{18}O < 3.25) (12)$$

$$T_d = 1 - 4.4 \times \frac{\delta^{18}O - 3.25}{3} \ IF \ (\delta^{18}O > 3.25) (13)$$


$T_s$ was then calculated for post-Pliocene using:

$$T_s = 2 \times T_d + 12.25 \ (14)$$

For the Pliocene:
$$T_s = 2.5 \times T_d + 12.15 \ (15)$$


And for pre-Pliocene we assumed that $T_s$ changed linearly with $T_d$, by a factor of 1.5.

$$\Delta T_s = 1.5 \times \Delta T_d \ (16)$$


$C_a$ based on an ESS range of 3–7°C was then calculated using the resulting $T_s$.

$$C_a = 310 \times \frac{T_{s[x]} - T_{s[0]}}{2 \times ESS} + 310 \quad (17)$$


In which $T_{s[x]}$ is the calculated average global surface temperature at time x, $T_{s[0]}$ is the modern day
average global surface temperature, and 310 represents pre-industrial $C_a$.

**3 Results and Discussion**

**3.1 Southern Temperate Rainforest Paleoecology**

Modern day Lauraceae rainforests in New Zealand have a single dominant canopy tree, *Beilschmiedia*

*tawa*, and its farthest southern extent is ~42°S (Leathwick, 2001), which is the farthest southern

occurrence of any arborescent Lauraceae species in the world. Rainforests further south in New Zealand

are usually dominated by Nothofagaceae or Podocarpaceae, and the only modern-day forests at ~50°S are

the Magellanic Subpolar Forests in southern South America. Low-growing Podocarpaceae/Nothofagaceae

forests, similar to modern forests in southern New Zealand and southern South America, dominated

Antarctic vegetation during the early Miocene (Askin and Raine, 2000) and the Foulden Maar rainforest

included at least ten Lauraceae species (Bannister et al., 2012), emphasizing the expanded biosphere

potential in the early Miocene compared to today (Herold et al., 2010).

        We identify *Litsea calicarioides*, *Cryptocarya maarensis*, *C. taieriensis*, cf.

Elaeocarpaceae/Cunoniaceae, and cf. Myrtaceae as the most probable canopy components because they

lack characteristics typical of understory components, 1) the large range of leaf $\delta^{13}C$ values and relatively

low overall leaf $\delta^{13}C$ values (Graham et al., 2014), 2) low cell densities (Kürschner, 1997; Bush et al.,

2017) (Fig. 2 a,b), and 3) the undulating or sinuous cell walls (Kürschner, 1997; Bush et al., 2017).

Modern day *Litsea calicaris* in New Zealand is also part of the canopy, though rarely dominant (de

Lange, 2020), whereas *Cryptocarya* is extinct in New Zealand. Members of Elaeocarpaceae, Cunoniaceae
and Myrtaceae in modern day New Zealand, such as *Weinmannia racemosa* (Cunoniaceae) and
*Metrosideros robusta* (Myrtaceae) can attain heights of over 25 meters (de Lange, 2020).
The most likely subcanopy or understory taxa were cf. *Ripogonum*, cf. *Myrsine*, "O", and cf.
*Dysoxylum*, because leaf fossils of these types have low overall leaf $\delta^{13}C$, relatively low cell densities, and
sinuous or undulating cells (Fig. S1e,f, S2b,d). *Ripogonum scandens* in modern day New Zealand is a
twining forest liana, often found in the understory, *Myrsine* comprises several species of shrubs and small
trees; whereas *Dysoxylum spectabile* in modern day New Zealand is a medium-sized tree (de Lange,
2020). The affinity of morphotype "O" is unclear, but likely represents a now extinct plant group in New
Zealand. *Hedycarya. pluvisilva*, *Laurelia. otagoensis*, cf. *Cryptocarya*, cf. *Endiandra*, *Beilschmiedia.*
*otagoensis*, "C", "H", "P", and "Q", all displayed some variation in features that are typical of understory
or canopy components and occurred in relatively low abundance, and are therefore considered of
uncertain ecological affinity.

**3.2 Earliest Miocene $CO_2$**
Gas-exchange modeling (Franks et al., 2014) of canopy leaves throughout the Foulden Maar core
indicates that $C_a$ (±1σ) was 445 +618 / -100 ppm, whereas reconstructed $C_a$ from understory elements
yields $C_a$ of 622 +3017 / -161 ppm (Fig. 2c), consistent with understory plants assimilating respired $CO_2$
that has undergone prior fractionation processes, as well as experiencing elevated levels of $C_a$ under the
canopy (Graham et al., 2014; Royer et al., 2017). Prior work on the Foulden Maar core established three
different phases based on bulk organic $\delta^{13}C$ (Fig. 1c), fatty acid $\delta^{13}C$, and fatty acid δD: Phase I (80–105
m depth) with high $\delta^{13}C$ and low δD, Phase II (55–65 m depth) with low $\delta^{13}C$ and high δD, and Phase III
(0–45 m depth) with high $\delta^{13}C$ and low δD (Reichgelt et al., 2016). Phase III can be further subdivided
into Phase IIIa (30–45 m depth) and IIIb (0–20 m depth), as Phase IIIa exhibits a period of low fatty acid
$\delta^{13}C$ and high δD, which is not expressed in bulk organic $\delta^{13}C$ (Reichgelt et al., 2016). Gas-exchange
modelling on leaves from these phases (Fig. 1c) suggest that during Phase II and IIIa $C_a$ may have been
elevated ($C_a$ = 529 +1159 / -125 and $C_a$ = 538 +769 / -181 ppm, respectively) compared to Phase I and
Phase IIIb ($C_a$ = 444 +572 / -95 and 442 +1219 / -110 ppm, respectively) (Fig. 3). Although gas-exchange
modeling input reconstructed differing $C_a$ between phases, differences in overall conductance parameters,
such as stomatal density and leaf $\delta^{13}C$, are not apparent (Table S1), despite differences in bulk $\delta^{13}C$, fatty
acid $\delta^{13}C$, and $\delta D$ (Reichgelt et al., 2016). This is likely the result of non-uniform species responses to
environmental changes in a complex multi-layered rainforest ecosystem, such as at Foulden Maar.

The advantage of using gas-exchange modeling to reconstruct $C_a$ from multiple species is that the

uncertainty is quantified and constrained, greatly reducing the potential for systematic error in the final
estimate (Reichgelt and D'Andrea, 2019; Royer et al., 2019). Along with the enhanced accuracy comes a
more comprehensive appraisal of uncertainty than is achieved using other proxy approaches (Fig. 4).
Proxy error propagation is based on mechanistic variability, grounded in known physical and
physiological limits of plant gas-exchange that are understood to be universal (Franks et al., 2014). This
differs from empirical proxies, whose uncertainty representation is based on calibration error of modern-
day observations without mechanistic constraints. Our canopy $C_a$ estimate (445 +618 / -100 ppm, Fig. 2c)
is independent of calibration error, based on universal gas-exchange mechanisms, and represents plant
vegetative organs of multiple plant species that directly interacted with the available pool of atmospheric
carbon dioxide. Previous $C_a$ estimates from the Oligocene/Miocene boundary based on boron isotopes
and paleosol carbonates are generally lower than our estimates (Ji et al., 2018; Greenop et al., 2019) (Fig.
4b), whereas $C_a$ estimates based on stomatal index and recent alkenone-based $C_a$ estimates are more
similar to our results (Kürschner et al., 2008; Super et al., 2018).

Reconstructions of globally elevated temperatures of 5–6 °C in the early Miocene (Hansen et al.,

2013) with a $C_a$ of ~300 ppm (Ji et al., 2018; Greenop et al., 2019) upsets the expected ESS to $C_a$ during
this period (Henrot et al., 2010). Geochemical $C_a$ proxy estimates consistently produce $C_a$ estimates that
are too low to satisfy ESS to $C_a$ prior to the Pliocene (Royer, 2016) (Fig. 4a,b). Estimates from the fossil
leaf-based stomatal index proxy for $C_a$ (Kürschner et al., 2008) on the other hand do indicate a positive
correlation between temperature and $C_a$ in the Neogene (Fig. 4a). At present, there are too few studies that
reconstruct $C_a$ using gas-exchange modeling to allow for a full comparison to other $C_a$ proxies; however,
our $C_a$ estimates of ~450–550 ppm are in line with the ESS to $C_a$ in the early Miocene (Fig. 4a,b), based
on modelling experiments (Herold et al., 2010; Henrot et al., 2010). Moreover, thus far, Neogene $C_a$
estimates reconstructed using gas-exchange methods (Reichgelt et al., 2016; Tesfamichael et al., 2017;
Londoño et al., 2018; Moraweck et al., 2019) appear to agree with the suggested ESS to $C_a$ (Fig. 4a,b).

Bulk organic and leaf wax $\delta^{13}C$ values reveal a ~4‰ decrease at Foulden Maar over a 10-meter

interval at the beginning of Phase II (55–65 m depth), likely representing a time period of <10 kyr (Fox et
al., 2016). This shift in isotopic composition suggests a substantial change in the global carbon cycle
(Reichgelt et al., 2016). The mode of reconstructed values in this study suggests and increase of ~450 to
550 from Phase I to Phase II (Fig. 3). The $C_a$ values stay near 550 ppm throughout Phase II and Phase
IIIa, representing a 20–40 kyr time period (Fig. 3). Absolute dating of Foulden Maar based on
paleomagnetic reversals in the core, annual lamination of lake sediments, and basalt-derived Ar/Ar dates
indicates that the deposition of the Foulden Maar sediment coincided with the termination of the earliest
Miocene (Mi-1) glaciation of Antarctica (Fox et al., 2015). Interestingly, an increase in $C_a$ from ~450 to
~550 ppm at the termination of Mi-1 is consistent with modeling studies indicating that $C_a > 500$ ppm is
necessary to terminate a large-scale Antarctic glaciation (DeConto et al., 2008). We note that with the
current data available, it is not possible to exclude the possibility that modeled $C_a$ changes in the record
were influenced by canopy density changes or regional hydroclimate. However, our observations from
Foulden Maar are inconsistent with hydrological, ecological or $C_a$ changes as the sole driver of plant
physiological response (Reichgelt et al., 2016), and it is more likely that two or more of these parameters
changed in concert.

**3.3 Elevated $CO_2$ and the early Miocene biosphere**
The Foulden Maar Miocene rainforest was primarily evergreen (Lee et al., 2016). The main Miocene
canopy trees at Foulden Maar, *Litsea calicarioides*, *Cryptocarya taieriensis*, *C. maarensis*, cf.
Elaeocarpaceae/Cunoniaceae and cf. Myrtaceae, had relatively high iWUE (Miocene iWUE first quartile
[$Q_1$] – third quartile [$Q_3$] = 70–101) compared to modern evergreen trees (evergreen iWUE $Q_1$–$Q_3$ = 31–
73) (Fig. 5a). Reconstructed iWUE from tropical early Miocene plants (Tesfamichael et al., 2017;
Londoño et al., 2018) is slightly higher ($Q_1$–$Q_3$ = 80–125) (Fig. 5a). The difference between reconstructed
Miocene iWUE and that of modern deciduous trees is greater still (deciduous iWUE $Q_1$–$Q_3$ = 27–52),
consistent with the expectation that increased $C_a$ favors evergreen trees (Niinemets et al., 2011; Soh et al.,
2019). In the method of reconstruction used here, iWUE is ultimately an expression of leaf $\delta^{13}C$ and
conductance (see Methods section 2.3). Therefore, similar to reconstructed $C_a$ iWUE may be sensitive to
environmental factors other than $C_a$. For example, leaf $\delta^{13}C$ can change in response to edaphic conditions
and precipitation (e.g. Kohn, 2016; Cornwell et al., 2018), as well as vapor pressure deficit (VPD) (Franks
et al., 2013). The climate of Foulden Maar in the early Miocene was warm-temperate to subtropical,
compared to the cool-temperate forests in southern New Zealand today (Reichgelt et al., 2019). Though
reconstructed relative humidity at Foulden Maar is within the range of modern New Zealand forest
biomes, the average monthly VPD was 500–700 Pa, compared to 250–450 Pa today (Fig. S3), which may
result in a similar reconstructed iWUE as under elevated $C_a$ (Franks et al., 2013). However, the
reconstructed ecosystem at Foulden Maar is a broad-leaved humid rainforest (Bannister et al., 2012),
which likely had a high annual moisture surplus (Reichgelt et al., 2019). Increased water-use efficiency in
response to relatively high VPD compared to modern would only be a positive trade-off if water
availability were limiting. Additionally, reconstructed iWUE from both temperate and tropical early
Miocene floras are high compared to modern, suggesting a global signal, such as would be expected to
globally elevated $C_a$; not from VPD as the early Miocene tropics would not be warmer than today (Herold
et al., 2010). Most importantly, the modern iWUE data (Fig. 5a) are from a global database that includes
environments with annual moisture deficits (Maire et al., 2015). Because reconstructed early Miocene
iWUE is higher even than modern plants experiencing high VPD, we argue that VPD differences cannot
explain the high iWUE values of the early Miocene, and that increased efficiency due to higher $C_a$ is the
best explanation.
In contrast to iWUE, reconstructed conductance to water ($G_w$) for Miocene trees is similar to the
modern-day range at the same latitude (Fig. 5b), a somewhat surprising result because $G_w$ is expected to
be reduced in high $C_a$ climates (Franks and Beerling, 2009). Studies on modern forests also suggest the
absence of a reduction in $G_w$ to enhanced $C_a$ (Yang et al., 2016; Gimeno et al., 2018), or even an increase
in the $G_w$ (Frank et al., 2015). A longer growing season together with increasing VPD was proposed to
explain increasing $G_w$ in modern European forests (Frank et al., 2015). Similarly, a relatively high water
flux from the forest to the atmosphere due to high water supply (Reichgelt et al., 2019) and high VPD
(Fig. S3) could explain the broad similarity in the range of modern and early Miocene $G_w$, despite higher
$C_a$. The early Miocene $G_w$ from tropical latitudes are within the range of modern evergreen tropical trees,
though relatively low (Early Miocene Q1–Q3: 0.08–0.13 mol $m^{-2}$ $s^{-1}$, modern evergreen Q1–Q3: 0.07–0.2
mol $m^{-2}$ $s^{-1}$) (Fig. 5b).
A longer growing season likely resulted in the high total annual carbon flux ($A_{tot}$) to the biosphere
reconstructed for Foulden Maar (Fig. 5c). Early Miocene trees at 50°S likely assimilated $A_{tot}$ $Q_1$–$Q_3$ =
265–696 gC $m^{-2}$ $yr^{-1}$, in comparison to $A_{tot}$ $Q_1$–$Q_3$ = 108–182 gC $m^{-2}$ $yr^{-1}$ in modern evergreen forests, and
$A_{tot}$ $Q_1$–$Q_3$ = 249–410 gC $m^{-2}$ $yr^{-1}$ in modern deciduous forests at the same latitude (Fig. 5c). Early
Miocene tropical trees appear to have slightly higher total annual carbon flux ($A_{tot}$ $Q_1$–$Q_3$ = 596–1220 gC
$m^{-2}$ $yr^{-1}$) than today ($A_{tot}$ $Q_1$–$Q_3$ = 329–721 gC $m^{-2}$ $yr^{-1}$), which, with a year-round growing season in the
early Miocene (like today), is likely attributable to a leaf-level fertilization effect, similar to what is
observed in modern carbon fertilization experiments (Norby et al., 2003; Bader et al., 2013; Yang et al.,
2016). Although this estimate cannot take the number of leaves per unit area into account, these results
suggest enhanced leaf-level productivity during higher than modern $C_a$ in the early Miocene.
The methods used in this study provide an alternate approach to controlled carbon fertilization
experiments, such as the Free Air Carbon Enrichment (FACE) experiments (e.g. Long et al., 2004),
toward investigating the effect of increased $C_a$ on the biosphere. FACE experiments provide data on the
physiological effects of carbon enrichment on species that evolved under, or had thousands of years to
adapt to pre-industrial $C_a$ ($\approx$ 280 ppm), and the physiological changes detected in canopy species are
measured as a direct response or over leaf generations (e.g. Norby et al., 2003; Yang et al., 2016). By
contrast, our data provide an insight into species that evolved under higher than pre-industrial $C_a$ and had
many generations of individuals to adapt to incrementally slow changes. The direct, or multi-year
physiological response of modern forest trees to enhanced $C_a$ is non-linear and non-uniform (e.g. Long et
al., 2004; Ainsworth and Long, 2005), and therefore further investigations into the physiology of ancient
plants operating in high-$CO_2$ worlds are needed to reveal the complexity of plant responses over
evolutionary timescales.

**4 Conclusions**
Leaf-level gas-exchange derived $C_a$ estimates suggest that early Miocene atmospheric $CO_2$ was higher
than pre-industrial levels at 450–550 ppm, further solidifying the growing consensus of relatively high
early Miocene global temperatures maintained by high atmospheric $CO_2$ (Kürschner et al., 2009;
Tesfamichael et al., 2017; Super et al., 2018; Londoño et al., 2018; Moraweck et al., 2019). A relatively
high $C_a$ in the early Miocene also satisfies an Earth System Sensitivity of 3–7°C (Hansen et al., 2013;
Royer, 2016). A potential shift in atmospheric $CO_2$ from 450 to 550, and back to 450, is recorded in the
100 kyr of sedimentation and leaf deposition at Foulden Maar. A disruption of the regional carbon and
hydrological cycle was also recorded in leaf-wax $\delta^{13}C$ and $\delta D$ (Reichgelt et al., 2016), and may be linked
to the Antarctic deglaciation at the termination of the Mi-1 (DeConto et al., 2008; Fox et al., 2015;
Liebrand et al., 2017).
The first record is provided of increased Miocene leaf-level intrinsic water-use efficiency in both
temperate New Zealand and the tropics, and we provide evidence for increased leaf-level productivity in
temperate New Zealand. Enhanced productivity and water-use efficiency on other landmasses in
temperate latitudes during the early Miocene, such as North America, Australia, and Asia, would have
had a major impact on the global carbon and water cycles. Our gas-exchange results from New Zealand,
supplemented with results from Ethiopia (Tesfamichael et al., 2017) and Panama (Londoño et al., 2018)
provide empirical evidence for high water-use efficiency in the globally warmer world of the early
Miocene, associated with elevated $C_a$. Tropical trees with high water-use efficiency compared to modern,
would have likely facilitated forest survival in climates where currently tropical savannas and grasslands
exist. These high water-use efficiency forests in the tropics likely persisted until the late Miocene when
reduced $C_a$ (Mejía et al., 2017) started favoring the expansion of grasslands, in particular grasslands with
the $C_4$ pathway that is more efficient under low $C_a$ and high temperatures (Strömberg, 2011; Polissar et
al., 2016).

Emission scenarios suggest that atmospheric $CO_2$ will reach our reconstructed early Miocene values of

450 ppm by 2030–2040 CE. While the global temperature response may lag the $C_a$ increase, and forest
habitat expansion is hampered by the slow dispersal and growth rate of climax forest trees and
anthropogenic influence (e.g., forest fragmentation and fire), early Miocene water-use efficiency and
productivity estimates provide insight into future-biosphere potential, as well as into selective pressures
that influence the types of plants that may proliferate under future elevated $C_a$.

**Acknowledgments.** We thank the Gibson family for kindly allowing us access to the site. Funding for
this research was provided by a Royal Society of New Zealand Marsden grant (UOO1115) to DEL, an
NSF grant (EAR13-49659) to WJD, a Vetlesen Foundation Climate Center grant to TR and WJD, and the
Lamont-Doherty Earth Observatory Summer Internship Program for Undergraduates awarded to ACVM.
Wei Huang, Andy Juhl and Nicole DeRoberts are acknowledged for technical support. Gregory Retallack
and an anonymous reviewer are acknowledged for their insightful comments that greatly improved this
manuscript in the review process.

**Author contributions.** TR and WJD conceived of the idea and performed data analyses. BRSF and DEL
collected sediment core, BRSF and TR sampled the sediment core, JGC and JMB identified fossil leaf
taxa. ACVM and TR gathered data from fossil leaves. TR and WJD wrote the paper and all authors
contributed to the final manuscript.

**Competing interests.** The authors declare no competing interests.

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

**Figure 1.** Foulden Maar site information. (a) Location of the Foulden Maar deposit and paleogeographic

reconstruction of early Miocene New Zealand (Boyden et al., 2011; Lee et al., 2014). (b) Schematic

reconstruction of the Foulden Maar depositional environment. (c) Stratigraphic column of the Foulden

Maar core (Fox et al., 2015), with sample locations and bulk organic $\delta^{13}C$ (Reichgelt et al., 2016).

Fig. 2

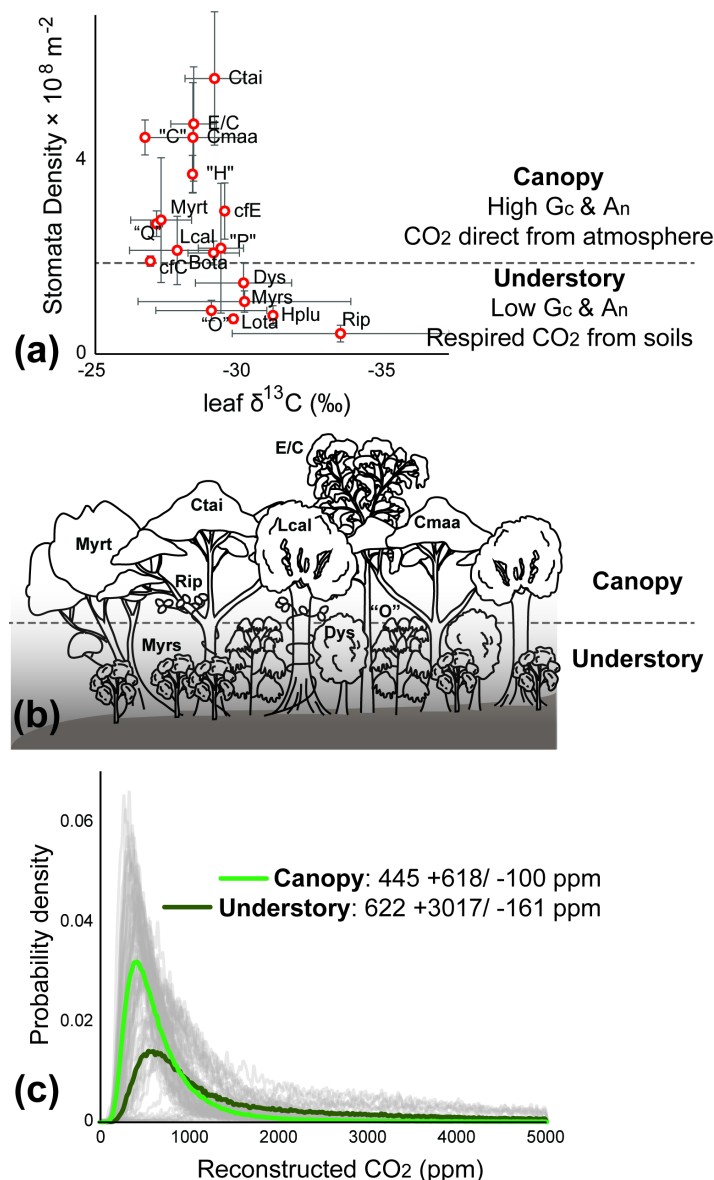

739

**Figure 2. Foulden Maar paleoecology and atmospheric CO₂ reconstructions. (a)** $\delta^{13}C$ values and stomatal density (error bars representing ±1σ) from the fossil leaves of Foulden Maar provide constraints to distinguish canopy leaf types from understory, because understory leaves tend to have a high range of $\delta^{13}C$ and low cell density (Graham et al., 2014; Bush et al., 2017). **(b)** Paleoecological reconstruction of the dense rainforest at Foulden Maar with a canopy comprising *Litsea calicarioides* (Lcal), *Cryptocarya taieriensis* (Ctai), *C. maarensis* (Cmaa), cf. Elaeocarpaceae/Cunoniaceae (E/C), and cf. Myrtaceae (Myrt), and an understory comprising cf. *Myrsine* (Myrs), cf. *Ripogonum* (Rip), cf. *Dysoxylum*

(Dys), and leaf type "O". *Hedycarya pluvisilva* (Hplu), *Laurelia otagoensis* (Lota), *Beilschmiedia*
*otagoensis* (Bota), cf. *Cryptocarya* (cfC), cf. *Endiandra* (cfE), and leaf types "C", "H", "P", and "Q"
could not be ecologically placed with certainty. **(c)** Probability density distributions of $C_a$ reconstructions
from canopy (thick light green line) and understory components (thick dark green line) using a gas-
exchange model (Franks et al., 2014). Grey curves represents the probability distribution of 10,000
Monte Carlo reconstructions on a single fossil leaf.

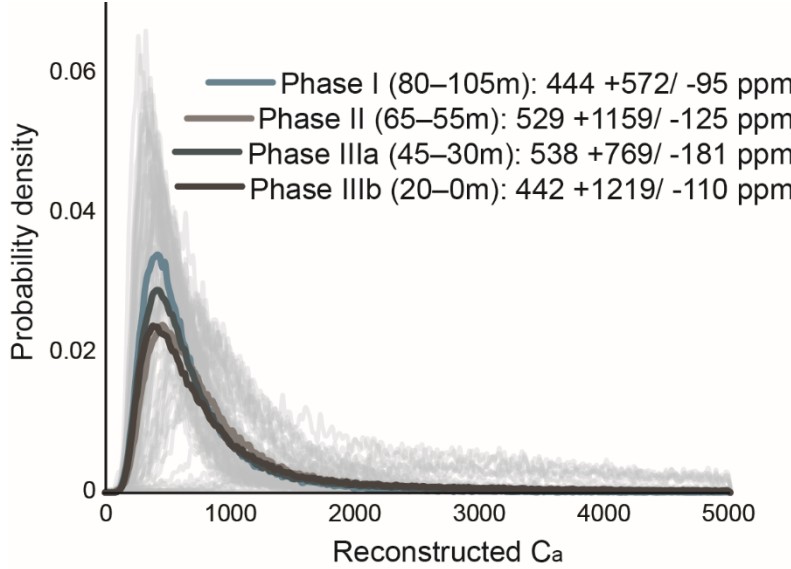


**Figure 3.** Probability density distributions of Ca reconstructions using a gas-exchange model (Franks et
al., 2014), divided by bulk carbon isotope phases (Fig. 1c).

Fig. 4

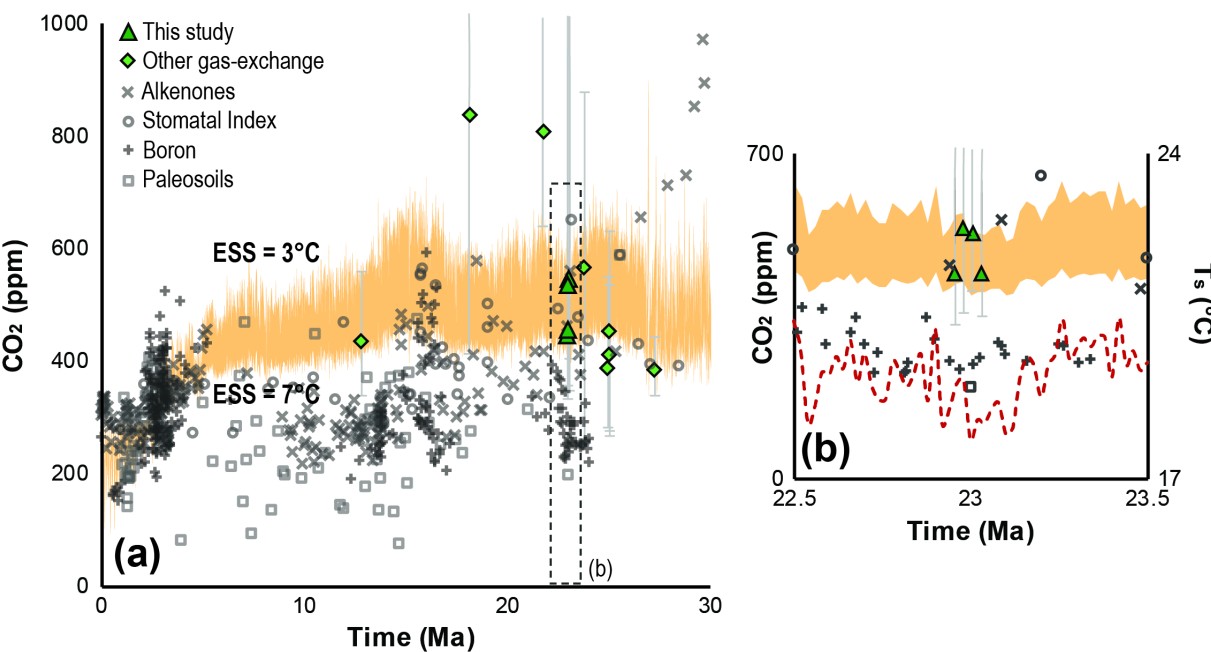


**Figure 4. Neogene Earth System Sensitivity (ESS) and C$_a$ reconstructions**. Calculated C$_a$ levels for an

ESS range of 3–7 °C (orange shaded area) for the last 30 Ma **(a)**, and for the interval between 22.5–23.5

Ma **(b)**, the red dashed line in **(b)** indicates the global average surface temperature (T$_s$) in the earliest

Miocene (Hansen et al., 2013). The ESS envelope was determined using deep-sea $\delta^{18}$O of benthic

foraminifera (Zachos et al., 2001) and the transform function approach from Hansen et al. (2013)

(Supplementary Information). Proxy-based Neogene C$_a$ reconstructions are derived from a previously

published compilation (Foster et al., 2017) and are supplemented with more recently published data (Ji

et al., 2019; Londoño et al., 2018; Super et al., 2018; Greenop et al., 2019; Moraweck et al., 2019,

Steinthorsdottir et al., 2019). Error bars on gas-exchange based proxy estimates represent ±1σ.

769

Fig. 5

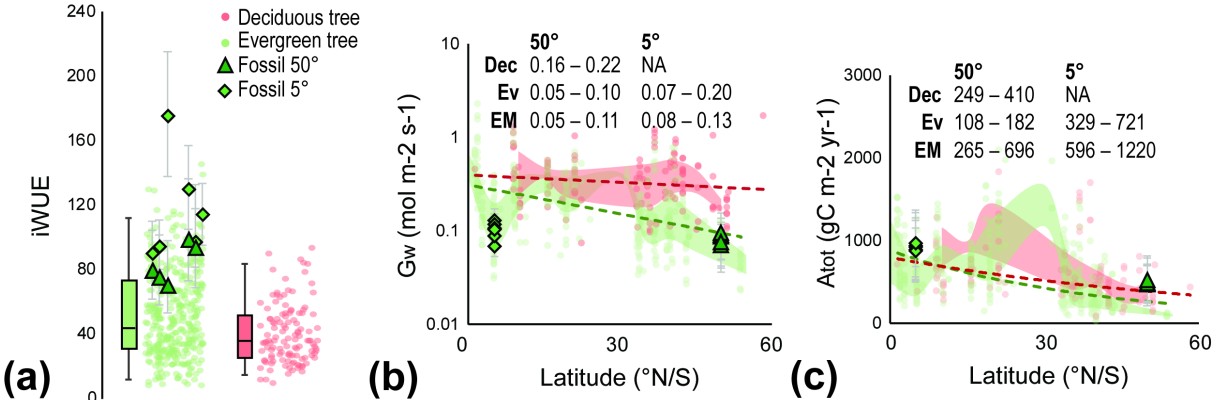

771

**Figure 5. Early Miocene leaf-level physiological parameters of canopy trees. (a)** Intrinsic water-use

efficiency (iWUE) of evergreen (green circles) and deciduous trees (red circles) based on modern leaf-

level measurements (Maire et al., 2015), and fossil reconstructions (green triangles and diamonds). Error

bars on fossil-derived data indicate $\pm1\sigma$, box-and-whisker plots indicate median, first and third quartile

(Q1 and Q3), and 95% confidence interval of modern leaves of canopy trees. Individual datapoints are

randomized on the x-axis for a clearer depiction of the distribution. **(b)** Conductance to water ($G_w$) from

modern evergreen (Ev) and deciduous (Dec) trees (Maire et al., 2015) and fossils (EM) from different

latitudes. The shaded red and green areas indicate the Q1–Q3 range of modern evergreen and deciduous

trees, respectively, and the dashed lines indicate the overall linear trend with latitude. Text in panel is the

Q1–Q3 range for each group, grouped in 5° latitude bins. **(c)** Total annual carbon flux per unit leaf area

($A_{tot}$) from modern evergreen (Ev) and deciduous (Dec) trees (Maire et al., 2015) and fossils (EM) from

different latitudes. The shaded red and green areas indicate the Q1–Q3 range of modern evergreen and

deciduous trees, respectively, and the dashed lines indicate the overall exponential trend with latitude.

Text in panel is the Q1–Q3 range for each group, grouped in 5° latitude bins.


**Data availability.** All raw measurement data on fossil leaves generated for this paper is available in the

online supplementary information. Raw measurements on fossil leaves from Ethiopia (Tesfamichael et

al., 2017) and Panama (Londoño et al., 2018), $\delta^{18}O$ measurements (Zachos et al., 2001), and iWUE, $G_w$

and $A_n$ measurements on modern plants (Maire et al., 2015) are available through the cited original
works.