# Peer review of "Elevated CO2, increased leaf-level productivity and water-use efficiency during the early Miocene"

_Climate of the Past, 2020_

## Referee Comment (RC1) · Anonymous Referee #1 · 6 May 2020

This manuscript is a timely contribution to an important topic, namely the question whether or not CO2 was elevated in the Miocene. Current data on Miocene temperature and CO2 are partially in conflict with various CO2 proxies indicating a much lower CO2 than necessary to explain Miocene climate. The authors represent CO2 reconstructions, based on fossil leaf material from the Foulden Maar, a well-studied site. There was already a wealth of material and data available which served as a valuable basis for this study. A disadvantage is that the considered sediments are dated to the earliest Miocene, very close to the Oligocene-Miocene boundary, and therefore comprise only a very limited time interval of the Miocene. CO2 was calculated with the model of Franks et al. (2014), in a competent manner. Methods and results are

sound and are a welcome contribution to the Miocene CO2 record. There are, however, various aspects which go beyond the general CO2 calculations and which are, in my opinion, not so well-founded and/or require additional discussions. These will be briefly explained in the following. 1) The authors try to erect a high-resolution sequence of CO2, for the different layers of the considered maar sediments. They found differences in CO2 calculated for these different layers and present these as proving CO2 fluctuations within the considered time interval. These fluctuations are not large (given as 450 to 550 ppm and back) considering the various uncertainties afflicting all CO2 proxy approaches. The Franks model is based on fossil stomatal data and fossil delta13C. Are the fluctuations in calculated CO2 caused by layer-specific differences in stomatal data or delta13C, or both? There is data scatter to be expected for both. It should therefore firstly be clarified whether or not the differences in stomatal data between layers are statistically significant. It would be also interesting to compare stomatal data of the fossil plants with those of their extant representatives. Are there significant differences? With respect to delta13C, there is also the problem of other environmental factors affecting this parameter, particularly humidity, as discussed further below.

2) A considerable topic of the study is the intrinsic WUE. This is the ratio between assimilation (here calculated with the Franks model) and stomatal conductance (derived from anatomical data from the fossil leaves and various assumed parameters). A further basic parameter of the Franks model is Ci/Ca (the ratio between internal and external CO2) which is derived on the basis of delta13C. Ci/Ca depends on both assimilation (thereby also on temperature) and stomatal conductance. In plant gas exchange regulation, humidity plays an essential role. Particularly, gs (stomatal conductance) tends to be lower under lower humidity and this is the main reason why 13C also carries a humidity signal which is influenced by other abiotic parameters in a complex way (see, for instance, Cornwell et al. 2018, Global Ecology and Biogeography). Furthermore, the "operational conductance" in the Franks model is based on an assumed aperture width, which does not take into account the regulation of stomatal aperture. To evaluate water

use efficiency on the basis of intrinsic WUE, information on paleoclimate is necessary. In this manuscript version, the treatment of WUE is too simplistic. In their 2016 publication on the Foulden Maar, distinct differences in wetness between the different layers of the considered sediment were inferred from deltaD isotopes by the authors. Therefore, evidence for environmental fluctuations already existed for the site and should be taken into account when aiming at identifying and discussing intrinsic WUE and also $CO_2$ fluctuations. I cannot understand why the authors did not make use of these former results. 3) In addition, the authors conclude from their results, particularly on the basis of an enhanced iWUE, a "general forest fertilization effect". It is generally difficult to extrapolate from leaf-level productivity to the canopy and vegetation level, and even more so for elevated $CO_2$, as is demonstrated by the variety of different observations on extant vegetation, including various FACE sites. Although there are various reports on "greening" of drier sites, the whole picture is much more complex. One question is, for example, whether or not leaf area per ground area (LAI) increases. There is evidence, that LAI does not increase in closed canopies (as was obviously the case for Miocene site considered in the presented study) under elevated $CO_2$, compared to ambient $CO_2$, when water is not (or not substantially) limited (Norby et al. 2003, Oecologia, 136. Yang et al. 2016 Journal of Geophysical Research: Biogeosciences, 121). Furthermore, with respect to forest water use under elevated $CO_2$, results are different for different forests (see for instance: Gimeno et al. (2018) Global change biology, 24, and Bader et al. (2013). Journal of Ecology, 101). Given the difficulties to pinpoint effects of elevated $CO_2$ for extant vegetation, it appears to be difficult to draw general conclusions for fossil plants. It is thus suggested that the authors mention and discuss this topic.

SPECIFIC COMMENTS P. 2, l. 42 "... will make more C available to the terrestrial biosphere..." This is an awkward description of the anticipated fertilization effect of elevated $CO_2$. P. 4, l. 98 "...For conductance measurements ..." This is not exactly correct. With fossil leaves, anatomical data are determined which then allow to approximate conductance (on the basis of various assumptions). This is not the same as

measuring conductance of living leaves. P. 4, l. 103 See previous comment. P. 8, ls. 194 – 195 There seems to be something wrong with the structure of this sentence. P. 10, l. 229 - 231 "...including a measure for the relative time the leaf is assimilating ..." What is the final value for this relative time? How was it determined? Additionally, the symbol for this relative time appears to be the same as for the operational conductance. P. 10, ls. 238 - 239 "...is derived from Maire et al. (2015) which included coordinates, habit, An and Gw data from which we could then calculate ..." It is not clear (from this sentence), how the calculations were conducted in detail. Why were "coordinates" used and for what? Why where Gw data from Maire et al. (and therefore of extant plants) used, and not conductance data derived from stomatal data of the fossil plants ? P. 15, ls. 355 – 357 "In contrast to iWUE...Gw for Miocene trees is similar to the modern day range ..." Since Gw is derived from Gc and therefore from fossil material, this would mean that "structural" conductance is not that different for the fossil plants and their extant relatives? P. 15, ls. 357 - 359 "Increased atmospheric evaporative demand in combination with a longer growing season ..." The authors describe that they used CLAMP to reconstruct growing season length. As far as I know, CLAMP provides also data on humidity. See also general comments.

———————————————

---

## Referee Comment (RC2) · Gregory Retallack (Referee) · 13 May 2020

This is a welcome contribution to CO2 paleobarometers and forest paleophysiology during a Miocene time of higher than current atmospheric CO2. The paper is thus relevant to understanding plant responses to currently rising atmospheric CO2.

This paper is well written and relatively free of errors, and presumably has been reviewed before. Nevertheless, three issues deserve further attention.

Earth System Sensitivity (l.242) is a very slippery concept in this context, because the temperature increase with CO2 doubling in any one part of the world will depend on

where it is. There are already numerous studies showing that midlatitude continental sites show little temperature change and thus muted sensitivity, but tropical and polar sites show marked changes in temperature. New Zealand is a temperate, site but also globally unusual in having a strongly marine-influenced climate now, and even more so in the Miocene when there was little land and few mountains. It is not clear how this even becomes relevant later (l.374) where elevated $CO_2$ estimated is thought to relate to ESS of 4-7oC, because Miocene paleotemperature for New Zealand is not offered.

I fail to see the relevance of C4 grassland expansion (l. 388) because it postdated the age of these New Zealand leaves by some 10 million years.

Errors in estimated paleoatmospheric $CO_2$ are asymmetric and very large (l. 307-8). Perhaps this is due to inadequate numbers of stomates counted: it should be hundreds in each image. Furthermore, Gaussian error propagation can be used to calculate symmetrical errors. Both issues are addressed in the following paper too recent to be included - Retallack, G.J. and Conde, G.D., 2020. Deep time perspective on rising atmospheric $CO_2$. Global and Planetary Change, p.103177.

---

## Author Comment (AC2) · 14 Jun 2020

Response to Referee #2, Gregory Retallack. Many thanks to Dr. Retallack for his insightful review. His comments are greatly appreciated.

Reviewer comment: This paper is well written and relatively free of errors, and presumably has been reviewed before.

Author response: It had not been reviewed before and we appreciate compliment.

Reviewer comment: Earth System Sensitivity (l.242) is a very slippery concept in this context, because the temperature increase with CO2 doubling in any one part of the

world will depend on where it is. There are already numerous studies showing that midlatitude continental sites show little temperature change and thus muted sensitivity, but tropical and polar sites show marked changes in temperature. New Zealand is a temperate, site but also globally unusual in having a strongly marine-influenced climate now, and even more so in the Miocene when there was little land and few mountains. It is not clear how this even becomes relevant later (l.374) where elevated CO2 estimated is thought to relate to ESS of 4-7oC, because Miocene paleotemperature for New Zealand is not offered.

Author response: Perhaps this wasn't clear enough in our discussion. We agree that local temperatures are not particularly instructive when considering ESS. The temperature change we considered was an estimate of the global average based on data and the transfer function approach presented in Hansen et al. (2013). We have updated the text to clarify this. While this approach itself has its limitations (which we also discuss in the manuscript), we believe that it is a reasonable basis for the ESS discussion and that this discussion is valuable.

Reviewer comment: I fail to see the relevance of C4 grassland expansion (l. 388) because it postdated the age of these New Zealand leaves by some 10 million years.

Author response: We updated the text to better explain the relevance of C4 grassland expansion in the late Miocene in the context of our results.

Reviewer comment: Errors in estimated paleoatmospheric CO2 are asymmetric and very large (l. 307-8). Perhaps this is due to inadequate numbers of stomates counted: it should be hundreds in each image. Furthermore, Gaussian error propagation can be used to calculate symmetrical errors. Both issues are addressed in the following paper too recent to be included - Retallack, G.J. and Conde, G.D., 2020. Deep time perspective on rising atmospheric CO2. Global and Planetary Change, p.103177.

Author response: The reviewer raises an interesting point about the numbers of stomata per image counted resulting in larger error bars. We have now added

text to further clarify how we avoided systematic error propagation and we now have referenced the reviewer's suggested study to highlight the importance of the number of counts per image. The asymmetrical error propagation is the result of the gas-exchange model iteratively solving for conductance, assimilation rate and carbon dioxide, centered around a representative assimilation rate (A0) measured under modern day atmospheric carbon dioxide. Because of this approach, the model returns a greater number of divergent solutions for fossil conductance, assimilation rate, and carbon dioxide for high $CO_2$ estimates, and fewer for low $CO_2$ estimates.

Please also note the supplement to this comment:
https://www.clim-past-discuss.net/cp-2020-30/cp-2020-30-AC2-supplement.pdf

---

## Author Response (AR1)

1 Response to Anonymous Referee #1. We thank the anonymous reviewer for an insightful and thoughtful 2 review. Below are our responses.

3

**4 Referee 1: Are fluctuations in calculated CO2 caused by layer-specific differences in stomatal data 5 or delta13C, or both? Are there significant differences in stomatal conductance or delta13C?**

6

7 Author response: Indeed, in the traditional approach to atmospheric CO2 reconstructions using changes in 8 plant physiology, inferred CO2 variations can be traced directly to either leaf carbon isotopic composition 9 or changes in stomatal density. However, the approach that we try to advocate relies on gas-exchange 10 modeling and it 1) is sensitive to any combination of changes in carbon isotopic composition and stomatal conductance, and 2) takes into account the cumulative response of different plant species (i.e. all the plant 11 12 species determined from the plant fossil locality). This approach thereby accounts for the complexities 13 that arise from non-linear, and even non-uniform physiological responses to changes in the climate, something that CO2 reconstructions using only stomata, or only leaf  $\delta^{13}$ C values, and only a single species 14 cannot do. Moreover, as we note in the manuscript this comprehensive approach leads to a more accurate 15

accounting of uncertainty in ultimate CO2 estimates than traditional approaches. 16

17

18 However, to address the reviewer's question, we conducted ANOVA linked with TukevHSD to test

differences in leaf  $\delta^{13}$ C and Stomatal Density between zones. We approached the ANOVA – TukeyHSD 19 20 with three different null hypotheses (H0): 1) leaf  $\delta^{13}$ C and stomatal density combining all species is the

same for all zones, 2) leaf  $\delta^{13}$ C and stomatal density for all canopy species, after Z-score scaling of inter-21

species variation, is the same for all zones, and 3) leaf  $\delta^{13}$ C and stomatal density for the most abundant 22

23 species, Litsea calicarioides, is the same for all zones. The p-value in all cases is higher than 0.05, indicating that H0 cannot be rejected in any of these scenarios, and that leaf  $\delta^{13}$ C and stomatal density do not 24

25 individually change significantly between zones. Thus, variations in estimated CO2 are the result of the

26 combination of leaf carbon isotopic composition, leaf conductance, and intra-species variation of

27 physiological response to atmospheric carbon. The original carbon isotope and leaf conductance

28 measurements are available in the supplementary material. We include new a section in the manuscript to

further clarify how our approach means that a change in model output may be impossible to trace to a 29

- uniform change in input variables, and on a related note we emphasized the need for further evidence to 30 31 further evaluate the role of a CO2 increase in driving Antarctic Ice melt at the Oligocene/Miocene
- 32 boundary.
- 33

**34 Referee 1: It would be also interesting to compare stomatal data of the fossil plants with those of 35 their extant representatives. Are there significant differences?**

36

37 Author response: We agree that this is an interesting research question, and it is currently considered in 38 the context of a separate study. The comparison between fossil plants and their extant representatives is 39 not of fundamental relevance to this manuscript and we prefer to keep it separate from the research results 40 we are reporting here.

41

**42 Referee 1: The treatment of intrinsic Water-Use Efficiency is too simplistic and should include 43 consideration of the changes in fatty acid $\delta D$ of the Foulden Maar record, in particular with**

44 regards to the influence of changes in humidity on plant water-use efficiency reconstructions.

45

Author response: We do have  $\delta D$  values and  $\delta^{13}C$  values from leaf waxes in this record that can provide 46

some guidance for making inferences about changes in hydroclimate across the 100,000-vr period of 47

sedimentation (Reichgelt et al., 2016). However, our discussion of iWUE is not meant to address 48

49 variations that occurred *during* this interval, but instead focuses on contrasting the early Miocene values

with modern values. To support our southern temperate reconstructed iWUE, we include results from the 50

same transform functions on previously published records from Ethiopia and Panama, which showed 51

| 52

58 | similar offsets from modern. That said, we agree with the referee that in a warmer world, whether you are
in the tropics or in the southern temperate region, you would expect higher vapor pressure deficits, which
would also drive up the iWUE signal. We have therefore expanded the discussion to address this
uncertainty and included Fig. S3 in the supplement to show that while temperatures Miocene New
Zealand are higher than modern, the relative humidity reconstructed for Foulden Maar is well within the
range of modern New Zealand forested biomes. |
|----------------------------------------|----------------------------------------------------------------------------------------------------------------------------------------------------------------------------------------------------------------------------------------------------------------------------------------------------------------------------------------------------------------------------------------------------------------------------------------------------------------------------------------------------------------------------------------------------------------------------------------|
| 59
60
61
62
63             | Referee 1: It is difficult to extrapolate leaf-level productivity to the canopy and vegetation level. It is suggested that the authors mention and discuss the research on modern CO2 fertilization experiments that highlight the complexity of physiological response in forests to increased atmospheric carbon dioxide.                                                                                                                                                                                                                                                            |
| 64
65
66                         | Author response: We expanded discussion on the confounding factors observed in modern CO2 fertilization experiments.                                                                                                                                                                                                                                                                                                                                                                                                                                                                   |
| 67
68
69                         | Referee 1: P. 2, l. 42 "will make more C available to the terrestrial biosphere". This is an awkward description of the anticipated fertilization effect of elevated CO2.                                                                                                                                                                                                                                                                                                                                                                                                              |
| 70
71                               | Author response: This sentence has been amended for clarity.                                                                                                                                                                                                                                                                                                                                                                                                                                                                                                                           |
| 72                                     | Referee 1: P. 4, l. 98 "For conductance measurements" This is not exactly correct. With fossil                                                                                                                                                                                                                                                                                                                                                                                                                                                                                         |
| 73
74                               | leaves, anatomical data are determined which then allow to approximate conductance (on the basis
of various assumptions). This is not the same as measuring conductance of living leaves P 4 1 103                                                                                                                                                                                                                                                                                                                                                                                  |
| 75
76                               | See previous comment.                                                                                                                                                                                                                                                                                                                                                                                                                                                                                                                                                                  |
| 77
78                               | Author response: amended.                                                                                                                                                                                                                                                                                                                                                                                                                                                                                                                                                              |
| 79
80                               | Referee 1: P. 8, ls. 194 – 195 There seems to be something wrong with the structure of this sentence.                                                                                                                                                                                                                                                                                                                                                                                                                                                                                  |
| 81
82                               | Author response: amended.                                                                                                                                                                                                                                                                                                                                                                                                                                                                                                                                                              |
| 83
84
85
86                   | Referee 1: P. 10, l. 229 - 231 "including a measure for the relative time the leaf is assimilating".
What is the final value for this relative time? How was it determined? Additionally, the symbol for
this relative time appears to be the same as for the operational conductance.                                                                                                                                                                                                                                                                                           |
| 87
88                               | Author response: amended.                                                                                                                                                                                                                                                                                                                                                                                                                                                                                                                                                              |
| 89
90
91
92
93             | Referee 1: P. 10, ls. 238 - 239 "is derived from Maire et al. (2015) which included coordinates, habit,
An and Gw data from which we could then calculate" It is not clear (from this sentence), how the
calculations were conducted in detail. Why were "coordinates" used and for what? Why where Gw
data from Maire et al. (and therefore of extant plants) used, and not conductance data derived from
stomatal data of the fossil plants?                                                                                                                             |
| 94
95
96                         | Author response: amended.                                                                                                                                                                                                                                                                                                                                                                                                                                                                                                                                                              |
| 97
98
99
100
101           | Referee 1: P. 15, ls. 355 – 357 "In contrast to iWUE Gw for Miocene trees is similar to the modern day range." Since Gw is derived from Gc and therefore from fossil material, this would mean that "structural" conductance is not that different for the fossil plants and their extant relatives?                                                                                                                                                                                                                                                                                   |

| 102
103
104
105
106 | Author response: That is correct. We have expanded the discussion on this. We note (here and in the manuscript) that the extant relatives are not the same as the plants that currently occur at this latitude. Due to cooling the warm-temperate to subtropical diverse Lauraceae dominated rainforests of Miocene New Zealand no longer exist. |
|---------------------------------|--------------------------------------------------------------------------------------------------------------------------------------------------------------------------------------------------------------------------------------------------------------------------------------------------------------------------------------------------|
| 107
108
109
110        | Referee 1: P. 15, ls. 357 - 359 "Increased atmospheric evaporative demand in combination with a longer growing season". The authors describe that they used CLAMP to reconstruct growing season length. As far as I know, CLAMP provides also data on humidity. See also general comments                                                        |
| 111                             | comments.                                                                                                                                                                                                                                                                                                                                        |
| 112                             | Author response: Thanks for this excellent suggestion CLAMP data on humidity have now been included                                                                                                                                                                                                                                              |
| 112
113
114               | in the supplementary material and are now included in our discussion.                                                                                                                                                                                                                                                                            |
| 115                             |                                                                                                                                                                                                                                                                                                                                                  |
| 116                             |                                                                                                                                                                                                                                                                                                                                                  |
| 117                             |                                                                                                                                                                                                                                                                                                                                                  |
| 118
119                      | Response to Referee #2, Gregory Retallack. Many thanks to Dr. Retallack for his insightful review. His comments are greatly appreciated.                                                                                                                                                                                                         |
| 120                             |                                                                                                                                                                                                                                                                                                                                                  |
| 121                             | Reviewer comment: This paper is well written and relatively free of errors, and presumably has                                                                                                                                                                                                                                                   |
| 122                             | been reviewed before.                                                                                                                                                                                                                                                                                                                            |
| 123                             |                                                                                                                                                                                                                                                                                                                                                  |
| 124                             | Author response: It had not been reviewed before and we appreciate compliment.                                                                                                                                                                                                                                                                   |
| 125                             |                                                                                                                                                                                                                                                                                                                                                  |
| 126                             | Reviewer comment: Earth System Sensitivity (1.242) is a very slippery concept in this context,                                                                                                                                                                                                                                                   |
| 127                             | because the temperature increase with CO2 doubling in any one part of the world will depend on                                                                                                                                                                                                                                                   |
| 128                             | where it is. There are already numerous studies showing that midlatitude continental sites show                                                                                                                                                                                                                                                  |
| 129                             | little temperature change and thus muted sensitivity, but tropical and polar sites show marked                                                                                                                                                                                                                                                   |
| 130                             | changes in temperature. New Zealand is a temperate, site but also globally unusual in having a                                                                                                                                                                                                                                                   |
| 131                             | strongly marine-influenced climate now, and even more so in the Miocene when there was little                                                                                                                                                                                                                                                    |
| 132                             | and and few mountains. It is not clear now this even becomes relevant later (1.5/4) where elevated                                                                                                                                                                                                                                               |
| 133                             | CO2 estimated is thought to relate to ESS of 4-/oC, because Miocene paleotemperature for New                                                                                                                                                                                                                                                     |
| 134                             | Zealand is not ollered.                                                                                                                                                                                                                                                                                                                          |
| 126                             | Author response: Perhaps this wasn't clear enough in our discussion. We agree that local temperatures are                                                                                                                                                                                                                                        |
| 130                             | not particularly instructive when considering ESS. The temperature change we considered was an                                                                                                                                                                                                                                                   |
| 138                             | estimate of the global average based on data and the transfer function approach presented in Hansen et al.                                                                                                                                                                                                                                       |
| 139                             | (2013) We have updated the text to clarify this While this approach itself has its limitations (which we                                                                                                                                                                                                                                         |
| 140                             | also discuss in the manuscript), we believe that it is a reasonable basis for the ESS discussion and that this                                                                                                                                                                                                                                   |
| 141                             | discussion is valuable.                                                                                                                                                                                                                                                                                                                          |
| 142                             |                                                                                                                                                                                                                                                                                                                                                  |
| 143                             | Reviewer comment: I fail to see the relevance of C4 grassland expansion (l. 388) because it                                                                                                                                                                                                                                                      |
| 144                             | postdated the age of these New Zealand leaves by some 10 million years.                                                                                                                                                                                                                                                                          |
| 145                             |                                                                                                                                                                                                                                                                                                                                                  |
| 146                             | Author response: We updated the text to better explain the relevance of C4 grassland expansion in the late                                                                                                                                                                                                                                       |
| 147                             | Miocene in the context of our results.                                                                                                                                                                                                                                                                                                           |
| 148                             |                                                                                                                                                                                                                                                                                                                                                  |
| 149                             | Reviewer comment: Errors in estimated paleoatmospheric CO2 are asymmetric and very large (l.                                                                                                                                                                                                                                                     |
| 150                             | 307-8). Perhaps this is due to inadequate numbers of stomates counted: it should be hundreds in                                                                                                                                                                                                                                                  |
| 151
152                      | each image. Furthermore, Gaussian error propagation can be used to calculate symmetrical errors.
Both issues are addressed in the following paper too recent to be included - Retallack, G.J. and                                                                                                                                             |

**Conde, G.D., 2020. Deep time perspective on rising atmospheric CO2. Global and Planetary Change, p.103177.**

154 155

156 Author response: The reviewer raises an interesting point about the numbers of stomata per image

- 157 counted resulting in larger error bars. We have now added text to further clarify how we avoided
- systematic error propagation and we now have referenced the reviewer's suggested study to highlight the
- 159 importance of the number of counts per image.
- 160 The asymmetrical error propagation is the result of the gas-exchange model iteratively solving for 161 conductance, assimilation rate and carbon dioxide, centered around a representative assimilation rate  $(A_0)$
- 162 measured under modern day atmospheric carbon dioxide. Because of this approach, the model returns a
- 163 greater number of divergent solutions for fossil conductance, assimilation rate, and carbon dioxide for 164 high CO2 estimates, and fewer for low CO2 estimates.
- 165
- 166
- 167
- 168

- 169 Elevated CO2, increased leaf-level productivity and water-use efficiency during the early Miocene
  170
- 171 Tammo Reichgelt1,2, William J. D'Andrea1, Ailín del C. Valdivia-McCarthy1, Bethany R.S. Fox3, Jennifer
- 172 M. Bannister4, John G. Conran5, William G. Lee6,7, Daphne E. Lee8
- 173

[revised manuscript text omitted]

operational conductance to Gmax (ζ), boundary layer conductance (Gb), and mesophyll conductance (Gm)
(Franks et al., 2014).

344

345
$$G_c = \left(\frac{1}{G_b} + \frac{1}{\zeta \times G_{max}} + \frac{1}{G_m}\right)^{-1} (4)$$

Gb,  $\zeta$ , and Gm are all changeable under natural conditions (e.g. Schuepp, 1993; Niinemets et al., 2009; 347 348 Londoño et al., 2018) and it is highly disputed if these variables can be determined from fossil leaf material at all (e.g. McElwain et a., 2016; Soh et al., 2017). However, we adopt a standardized approach 349 put forward by Franks et al. (2014) to obtain input for these variables.  $G_b = 2 \pm 0.1 \text{ mol m}^{-2} \text{ s}^{-1}$ ,  $\zeta = 0.2 \pm 0.1 \text{ mol m}^{-2} \text{ s}^{-1}$ 350 0.02 (Franks et al., 2009; Dow et al., 2014), and Gm 
[revised manuscript text omitted]

  791–795, 2016.
- 895